# Impact of the COVID-19 Pandemic on Anti-Vascular Endothelial Growth Factor Therapy for Diabetic Macular Edema in Japan

**DOI:** 10.3390/jcm11226794

**Published:** 2022-11-16

**Authors:** Ryohei Komori, Yoshihiro Takamura, Yutaka Yamada, Masakazu Morioka, Hisashi Matsubara, Takao Hirano, Yoshinori Mitamura, Miho Shimizu, Sentaro Kusuhara, Tomoya Murakami, Ryotaro Nihei, Tetsuo Ueda, Hiroshi Kunikata, Tatsuya Jujo, Hiroto Terasaki, Daisuke Nagasato, Kousuke Noda, Rie Osaka, Kazuki Nagai, Shigeo Yoshida, Miho Nozaki, Hidetaka Noma, Gaku Ishigooka, Aya Takahashi, Osamu Sawada, Kazuhiro Kimura, Masaru Inatani

**Affiliations:** 1J-CREST (Japan Clinical REtina STudy Group), Kagoshima 890-8544, Japan; 2Department of Ophthalmology, Faculty of Medical Sciences, University of Fukui, Yoshida 910-1193, Japan; 3Department of Ophthalmology, Mie University Graduate School of Medicine, Tsu 514-8507, Japan; 4Department of Ophthalmology, Shinshu University School of Medicine, Matsumoto 390-8621, Japan; 5Department of Ophthalmology, Tokushima University Graduate School, Tokushima 770-8503, Japan; 6Department of Ophthalmology, Sapporo City General Hospital, Sapporo 060-8604, Japan; 7Division of Ophthalmology, Department of Surgery, Kobe University Graduate School of Medicine, Kobe 650-0017, Japan; 8Department of Ophthalmology, Faculty of Medicine, University of Tsukuba, Tsukuba 305-8576, Japan; 9Department of Ophthalmology, National Defense Medical College, Tokorozawa 359-8513, Japan; 10Department of Ophthalmology, Nara Medical University, Kashihara 634-8522, Japan; 11Department of Ophthalmology, Tohoku University Graduate School of Medicine, Sendai 980-8574, Japan; 12Department of Ophthalmology, St Marianna University School of Medicine, Kawasaki 216-8511, Japan; 13Department of Ophthalmology, Kagoshima University Graduate School of Medical and Dental Sciences, Kagoshima 890-8544, Japan; 14Department of Ophthalmology, Saneikai Tsukazaki Hospital, Himeji 671-1227, Japan; 15Department of Ophthalmology, Faculty of Medicine and Graduate School of Medicine, Hokkaido University, Sapporo 060-8638, Japan; 16Department of Ophthalmology, Kagawa University Hospital, Kita 761-0793, Japan; 17Department of Ophthalmology, Gunma University Hospital, Maebashi 371-8511, Japan; 18Department of Ophthalmology, Faculty of Medicine, Kurume University, Kurume 830-0011, Japan; 19Department of Ophthalmology and Visual Science, Nagoya City University Graduate School of Medical Sciences, Nagoya 467-8601, Japan; 20Department of Ophthalmology, Tokyo Medical University Hachioji Medical Center, Tokyo 193-0998, Japan; 21Department of Ophthalmology, Osaka Medical and Pharmaceutical University, Takatsuki 569-8686, Japan; 22Department of Ophthalmology, Faculty of Medicine, Kyorin University, Mitaka 181-8611, Japan; 23Department of Ophthalmology, Shiga University of Medical Science, Otsu 520-2192, Japan; 24Department of Ophthalmology, Yamaguchi University School of Medicine, Ube 755-0046, Japan

**Keywords:** anti-VEGF therapy, COVID-19, diabetic macular edema

## Abstract

Anti-vascular endothelial growth factor (VEGF) therapy for diabetic macular edema (DME) improves visual acuity. However, repeated injections during routine outpatient visits are required to maintain this effect. The recent sudden global outbreak of coronavirus disease 2019 (COVID-19) had a major impact on daily life, including medical care, such as the provision of VEGF therapy. We retrospectively investigated the relationship between the number of anti-VEGF injections for DME and the number of new COVID-19-positive patients at 23 centers in Japan. We also surveyed ophthalmologists regarding the impact of the COVID-19 pandemic on anti-VEGF therapy. In the third and fourth waves of the pandemic, when the number of infected patients increased, the number of injections significantly decreased. In the first, third, and fourth waves, the number of injections increased significantly during the last month of each wave. Approximately 60.9% of ophthalmologists reported that the number of injections decreased after the pandemic. Of the facilities, 52.2% extended the clinic visit intervals; however, there was no significant difference in the actual number of injections given between before and after the pandemic. Although the number of injections temporarily decreased, Japanese ophthalmologists maintained the total annual number of anti-VEGF injections for DME during the pandemic.

## 1. Introduction

Diabetic macular edema (DME) is the main ocular disease that impairs central vision in patients with diabetic retinopathy [1]. DME is caused by increased permeability of retinal capillaries, microaneurysms, and disruption of the blood–retinal barrier [2]. Prolonged hyperglycemia leads to chronic retinal microvascular damage and hypoxia, and increases intraocular levels of vascular endothelial growth factor (VEGF) [2,3]. At present, intravitreal injection of anti-VEGF agents is the first-line treatment for DME [4]. Many large-scale clinical studies have demonstrated that this therapy has a promising effect in normalizing retinal thickness and improving visual acuity [5,6]. Anti-VEGF treatment regimens include monthly and bi-monthly dosing, as well as the pro re nata (PRN) regimen, in which injections of anti-VEGF agents are administered in response to the recurrence of the edema, and the treat and extend (TAE) regimen, in which the injection interval is coordinated in accordance with the retinal thickness and visual acuity [7].

A study using an animal model showed that, after a single injection of anti-VEGF agent, the concentration of the agent in the vitreous decreased over time [8,9]. Indeed, frequent injections of anti-VEGF agents are required to maintain the therapeutic effect. Matsunaga et al. showed that anti-VEGF-treated patients who were lost to follow-up for a prolonged period experienced a modest decline in visual acuity, which recovered after restarting injections [10]. In addition, we recently reported that better visual improvement cannot be achieved without prompt injection once edema recurrence has been identified [11]. Therefore, it is clinically important to monitor the status of DME through regular visits to ophthalmologists, and to administer anti-VEGF agents at the appropriate time.

Coronavirus disease 2019 (COVID-19) was first reported in China in December 2019 and was detected in Japan in January 2020. The number of COVID-19 patients increased dramatically, peaked, and then decreased, repeatedly. To date (July 2022), six such waves have been observed in Japan [12]. The outbreak of the COVID-19 pandemic has had a major impact on human activities, including medical care. Outpatient and surgical volumes decreased during the height of the pandemic, and treatment was restricted to urgent or emergency conditions [13,14]. The progression of COVID-19 can lead to acute respiratory distress syndrome and even death, and its common symptoms are fever, cough, and fatigue [15]. In ophthalmology practice, the distance between the ophthalmologist and patient is small. Although Ophthalmology Departments often receive patients without symptoms of infection, it is fundamental to follow the rules of infection control to prevent nosocomial infections [16]. Ophthalmologists are thus instructed to wear gloves and masks and attach a shield to the slit-lamp microscope [16]. While it may not have affected the number of injections of anti-VEGF agents, the pandemic also affected the way of practice in ophthalmology clinic.

In this multicenter study in Japan, we conducted a questionnaire-based survey to determine how ophthalmologists responded to the COVID-19 pandemic. In addition, we investigated the relationship between the number of COVID-19 positive patients and the number of anti-VEGF injections in patients with DME.

## 2. Methods

This multicenter, retrospective, observational study was conducted at 23 institutions in Japan. This study was approved by the University of Fukui Institutional Review Board (IRB No. 20210144) and the ethics committees of the other participating facilities: Mie University, Shinshu University, Tokushima University, Sapporo Municipal Hospital, Kobe University, Tsukuba University, National Defense Medical College, Nara Medical University, Tohoku University, St. Marianna Medical University, Kagoshima University, Tsukazaki Hospital, Hokkaido University, Kagawa University, Gunma University, Kurume University, Nagoya City University, Tokyo Medical University, Hachioji Medical Center, Osaka Medical and Pharmaceutical University, Kyorin University, Shiga Medical University, and Yamaguchi University. The study complied with the principles of the Declaration of Helsinki for all research procedures. It was registered in the UMIN Registry (Number 000048935). No personal identifiable information was collected or stored, and thus informed consent was waived. For COVID-19 positive patients, inclusion of visual acuity or central retinal thickness in the data was not permitted.

We tabulated the number of intravitreal injections of aflibercept and ranibizumab for DME administered by month from the records of the 23 institutions, from January 1, 2019 to February 28, 2022. We also tallied the number of new COVID-19 infections by month for the period from February 1, 2020, to February 28, 2022, based on the number of cases released by each prefecture where the participating institutions are located. The period from the first to the sixth waves was defined based on the number of new COVID-19-positive patients released by the Japanese Ministry of Health, Labor, and Welfare. The number of intravitreal injections of anti-VEGF drugs administered in patients diagnosed with DME per month was measured over the same period.

Intravitreal injections were performed in a standard manner by a trained ophthalmologist using 0.4% oxybuprocaine hydrochloride (0.4% Benoxil ophthalmic solution, Santen Co. Ltd., Osaka, Japan) and 2% xylocaine as anesthetic and povidone iodine for sterilization. The injection volumes of ranibizumab (Lucentis; Novartis Pharma K.K., Tokyo, Japan) and aflibercept (Eylea; Bayer Yakuhin Ltd., Tokyo, Japan) was 0.5 mg/0.05 mL and 2 mg/0.05 mL, respectively.

Additionally, a questionnaire survey with four items was administered to the chief of the ophthalmologists’ group treating DME at each facility. These items were as follows: (1) Do you feel that the number of anti-VEGF injections and number of outpatient visits have changed as compared to pre-pandemic levels? (2) For what period of time before and after COVID-19 vaccination were intravitreal injections not administered? (3) Did you extend the interval between patients’ visits to the ophthalmologist due to the COVID-19 pandemic? (4) Did you change the anti-VEGF treatment regimen for DME? We also asked if the ophthalmologists used masks, gloves, and slit-lamp microscope shields during outpatient consultation since the start of the pandemic. The questionnaire survey was conducted in March 2022.

JMP 10.0.2 (SAS Institute Japan Ltd., Tokyo, Japan) was used for statistical analysis. A paired *t*-test was used to compare between the different time points. All data are expressed as mean ± SE. The level of statistical significance was set at *p* < 0.05.

## 3. Results

### 3.1. Temporal Profiles of the Number of the COVID-19 Positive Patients and Injections

The average monthly numbers of new COVID-19-positive patients and of intravitreal injections of anti-VEGF drugs in DME patients are shown in Figure 1. During the observational period, six waves of COVID-19 occurred, as shown by the temporal profiles of the number of newly diagnosed COVID-19-positive patients. In the first month of the third wave of the pandemic, when the number of COVID-19 positive patients began to increase (October 2020 to November 2020), the number of anti-VEGF drug injections decreased significantly (*p* = 0.0218). During the increasing phase of the fourth wave (March 2021 to May 2021), the number of anti-VEGF injections again decreased significantly (*p* = 0.0014). In contrast, the number of injections increased significantly during the last month of the convergence of waves 1 (*p* = 0.0195), 3 (*p* = 0.0345), and 4 (*p* = 0.0023). No significant monthly change in the number of injections administered was observed during waves 2, 5, and 6. No significant differences in the monthly average number of injections of anti-VEGF drugs was found among first to sixth waves (*p* > 0.05, comparison of all pairs using Turkey–Kramer’s HSD test, wave 1: 25.39 ± 5.08, wave 2: 27.66 ± 5.01, wave 3: 27.26 ± 5.29, wave 4: 27.08 ± 4.96, wave 5: 28.79 ± 5.07, wave 6: 29.24 ± 5.12). The annual number of injections in 2019, before the pandemic, was 315.63 ± 58.85, while those in 2020 and 2021 were 316.84 ± 59.97 and 330.95 ± 58.37, respectively. The monthly number of injections in 2019, before the pandemic, was 26.30 ± 4.90, while those in 2020 and 2021 were 26.40 ± 5.00 and 27.58 ± 4.86, respectively. No significant differences were observed between the annual numbers of injections (*p* > 0.05, comparison of all pairs using Turkey–Kramer’s HSD test). In this study, none of the patients who received anti-VEGF therapy died from COVID-19.

### 3.2. Questionnaire

The survey responses obtained from the 23 facilities are shown in Figure 2. Compared to the period before the COVID-19 pandemic occurred, 60.9% and 87.0% of facilities considered the number of injections Figure 2A(a) and outpatients Figure 2A(b) decreased in the first year of the pandemic (1 February 2020 through 31 January 2021: corresponding to waves 1–3). Fewer facilities felt that the number of injections (39.1%) and patients (65.2%) had decreased in the second year (1 February 2021 through 27 February 2022: corresponding to waves 4–6) than did in the first year.

When asked how long to avoid injecting anti-VEGF drugs before vaccination, 4.3%, 13.0%, and 21.7% responded 1 day, 3 days, and 1 week, respectively (Figure 2B). The percentages of facilities that responded that the period of time during which injections were not administered after vaccination was 1 day, 3 days, 1 week, and 2 weeks were 13.0%, 30.4%, 21.7%, and 8.7%, respectively. Prior to the pandemic, the anti-VEGF treatment regimen at all facilities was PRN. Twelve facilities (52.2%) reported that they extended the interval between visits of the DME patients to the clinic Figure 2C(a). Four facilities (17.4%) changed the mode of treatment from PRN to TAE during the pandemic Figure 2C(b).

To prevent infection during outpatient consultations, ophthalmologists at all facilities wore masks and attached shields to the slit-lamp microscopes. Of the facilities, 30.4% additionally fitted staff with face guards. In 22 facilities (95.7%), the ophthalmologists used alcohol disinfection of hands, and at 14 facilities (60.9%) they used plastic gloves.

## 4. Discussion

Intravitreal injection of anti-VEGF agents is currently the first-line treatment for DME [4,17]. According to a survey of Japanese retinal specialists, the most common anti-VEGF treatment regimen is PRN, which involves injection at each recurrence [7,18]. However, delayed injection timing for DME recurrence has been reported to worsen visual prognosis [11]. Moreover, an increase in drop-offs and interruptions in ophthalmology visits are associated with worse visual prognosis [19]. Therefore, monitoring DME by regular visits is essential for maintaining a favorable visual improvement effect with anti-VEGF therapy. In this study, we investigated the influence of the COVID-19 pandemic on the number of anti-VEGF injections administered for DME. The COVID-19 pandemic has had a major impact on ophthalmic care in Japan. The COVID-19 pandemic has been reported to have reduced the rate of hospital visits greatly [16], possibly because both patients and healthcare workers were concerned about infection. Additionally, ophthalmology guidance was issued to prioritize urgent surgeries [16]. The results of the survey conducted in the present study showed that, during the year since the pandemic began, 87.0% of the participating facilities responded that the number of outpatients had decreased, and that 52.2% of facilities had extended the interval between clinic visits after the start of the pandemic. It is possible that social restrictions and inadequate number of visits on clinic may be contributing to poor glycemic control in the patients with diabetes.

In Japan, there have been at least six waves of COVID-19, based on the numbers of positive-testing patients during the observation period since its first identification in January 2020 until February 2022. Our analysis showed that, during the third and fourth waves of the pandemic, an increase in the number of patients and a decrease in the number of anti-VEGF drug injections occurred simultaneously. There are two possible reasons for the confinement of this inverse correlation between the number of patients and the number of injections to the third and fourth waves. In the first and second waves of the pandemic, the spread of infection was limited to urban areas. In the third and fourth waves, the number of COVID-19-positive patients increased nationwide, which may have caused patients to refrain from visiting their ophthalmologists, as reflected in the decrease in the number of injections. The second reason is the reassurance provided by the widespread use of vaccination. The widespread use of vaccination, considered an effective medical treatment against COVID-19, began with older individuals and healthcare workers during the third and fourth waves of the pandemic [20]. Hence, in the fifth wave, the number of anti-VEGF agent injections was probably less affected by the number of infected patients, because of a sense of relief that the number of infected patients had decreased due to the widespread use of vaccines. Behavioral restrictions implemented to prevent infection, such as the use of masks and face guards and maintaining a safe distance from others, were also established during this period. In addition to the lack of established knowledge about COVID-19 and mechanisms to prevent infection, lockdown of the city and reduced income due to reduced economic activity also increased people’s anxiety. The inverse correlation between the increase in the number of COVID-19-positive patients and anti-VEGF treatment may therefore reflect a variety of physical, emotional, and economic anxieties regarding COVID-19.

During the third and fourth waves, the number of injections decreased as the number of COVID-19 positive patients increased; however, the average number of injections during these periods was not significantly different from that in the first, second, fifth, and sixth waves. One possible reason for this is that the number of injections increased significantly with the decrease in the number of positive patients in each of the 1-month periods at the end of waves 1, 3, and 4. It can be assumed that the number of patients who received injections had increased due to the sense of relief resulting from a lightening of the pandemic. In other words, during the convergence period, many injections were administered to compensate for patients who refrained from visiting the ophthalmologists during the pandemic. Since the 6th wave did not subside at the endpoint of this study, it is unclear in the relationship between the number of the injections and the COVID-19 positive patients in the convergence period of the 6th wave and thereafter. Although the monthly number of injections did not differ significantly among the pandemic waves, many ophthalmologists (60.9%) indicated in the questionnaire that they felt that the number of injections had decreased since the start of the pandemic. This discrepancy may be because of the strong impression left by the decreased number of injections during the third and fourth waves. The first and second years after the start of the pandemic corresponded to waves 1–3 and waves 4–6, respectively. The survey results indicated that fewer facilities felt that there was a pandemic-induced decrease in injections and outpatient visits in the second year than in the first year. In the second year, the number of infected patients increased markedly as compared to that in the first year, but the risk of serious illness decreased due to the mutation of the virus and the widespread use of vaccines. The anxiety of both patients and ophthalmologists seems to have reduced and medical activity appears to have returned to pre-pandemic levels.

In the early days of the COVID-19 pandemic, information on the effectiveness and side effects of vaccines against COVID-19 is scarce. The effect of the vaccines on intravitreal anti-VEGF injections is not well-understood. Thus, this survey found that the timing of injections in relation to vaccine administration varied across facilities. Of the facilities, 73.8% reported that they did not administer anti-VEGF injections immediately after vaccination. Injections for DME were considered less urgent than were treatment for other ocular diseases, such as retinal detachment and endophthalmitis, and therefore were easier to postpone. In fact, 52.2% of the facilities surveyed indicated that they had extended the interval between visits for injections. Nevertheless, the actual number of injections per year did not show a significant decrease as compared with the pre-pandemic levels. One possible reason is that the number of injections did not decrease because of the larger number of injections administered at the end of COVID-19 waves, even though the interval between visits had increased. Alternatively, the change from the PRN to the TAE regimen might reduce the number of clinic visits because of the planned injections. However, Japanese ophthalmologists generally made efforts to avoid changing their pre-pandemic anti-VEGF treatment policies and injection numbers.

To minimize exposure to COVID-19, the guidelines recommend that ophthalmologists should wear gloves and masks, limit unnecessary conversations and examinations, and keep patients at a distance from each other in waiting rooms [16]. Our questionnaire showed that ophthalmologists at all facilities wore masks and attached shields to the slit-lamp microscopes. Additionally, face guards and gloves were used to protect the ophthalmologists against infection. These findings suggest that Japanese ophthalmologists should follow these guidelines and make efforts to prevent infection.

This study had several limitations. First, the participating facilities were limited to university hospitals and do not reflect data from other clinics. It is ideal to be expanded all clinics around Japan. Changes in the number of COVID-19-positive patients at each participating facility could not be examined, to protect personal information. Therefore, the number of patients released for the entire prefecture in which the facility was located was used in the data analysis. However, patients themselves only knew the number of COVID-19-positive patients released by the prefecture, which is the number thought to have influenced patients’ psychological status and their visits to medical facilities. As a second limitation, in this study we could not compare the changes in central retinal thickness or visual acuity before and after the pandemic. The reason for this is that COVID-19 positive individuals may have been included in our data. To prevent disclosure of personal information, we were not able to examine the visual acuity or central retinal thickness of the COVID-19 positive patients. On the other hand, the injection number of anti-VEGF agents could be examined regardless of the infection of COVID-19. Thus, it is unclear what effect the pandemic had on the improvement in visual acuity and edema in this study.

In conclusion, this study showed that, although there was a temporary decrease in the number of anti-VEGF injections during the third and fourth waves of the COVID-19 pandemic, the annual number of injections for the treatment of DME was in fact maintained at a level similar to that before the pandemic. Moreover, Japanese ophthalmologists followed recommended precautions to prevent infection.

## Figures and Tables

**Figure 1 jcm-11-06794-f001:**
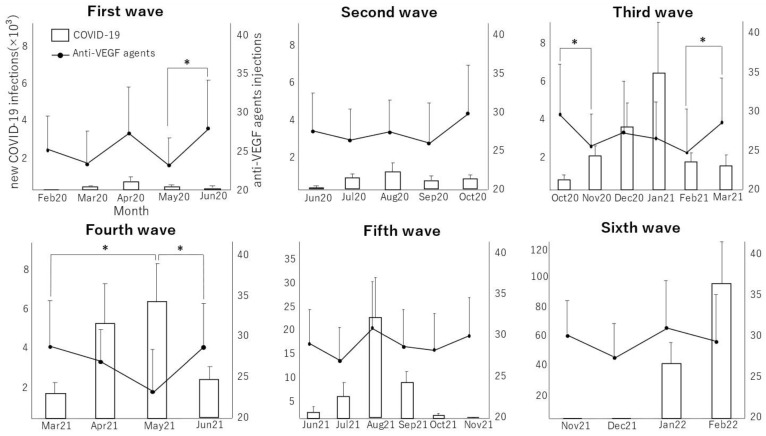
Changes in the number of new COVID-19-positive patients and in the number of injections of anti-VEGF agents during the waves of the COVID-19 pandemic. The data are shown as mean ± standard error. * *p* < 0.05.

**Figure 2 jcm-11-06794-f002:**
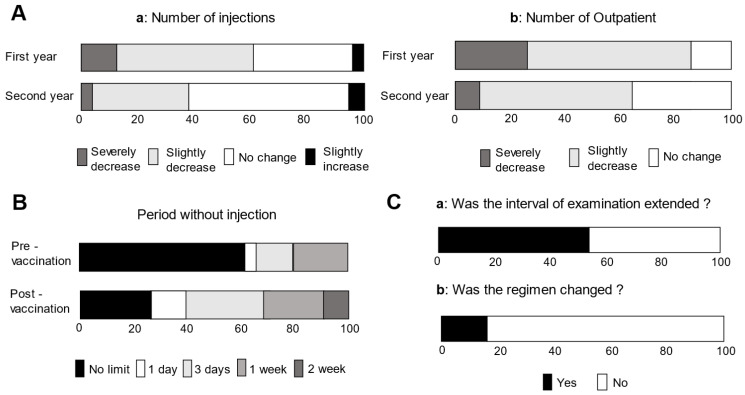
Distribution of responses to the questionnaires. (**A**) Responses to the question “Do you feel that the number of anti-VEGF injection (**a**) and number of outpatient visits (**b**) have changed as compared to pre-pandemic levels?” (**B**) Responses to the question “For what period of time before and after COVID-19 vaccination were intravitreal injections not administered?” (**C**) (**a**) Responses to the question “Did you extend the interval between patients’ visits to the ophthalmologist due to the COVID-19 pandemic?” and (**b**) Responses to the question “(4) Did you change the anti-VEGF treatment regimen for DME?”.

## Data Availability

The datasets generated and/or analyzed during the current study are available from the corresponding author upon reasonable request.

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
