# Peer review of "Impact of the COVID-19 Pandemic on Anti-Vascular Endothelial Growth Factor Therapy for Diabetic Macular Edema in Japan"

_jcm, 2022, doi:10.3390/jcm11226794_

Round 1

Reviewer 1 Report

This study retrospectively investigated the relationship between the number of anti-VEGF injections for DME and the number of new COVID-19-positive patients at 23 centers during the six waves of the COVID-19 pandemic from January 1, 2019, to February 28, 2022, in Japan. Authors found that the anti-VEGF treatment decreased when the number of infected patients increased during the 3rd and 4th waves of the pandemic. The number of new COVID-19 cases lessened in the 5th and 6th waves of pandemic, possibly because of the widespread vaccination.

The questionnaire-based survey showed that ophthalmologists felt the COVID-19 pandemic was the cause of the reduction of anti-VEGF treatment for DME in Japan. Nevertheless, the total number of anti-VEGF treatment was not different between the years before and after the onset of COVID-19 pandemic. Authors deduced that the number of anti-VEGF treatment may be increased during the peak of infection of each wave of the pandemic.

Comments

1.     The mean number of anti-VEGF injections per month ranged from 25 to 29 during the six waves of pandemic. What is the mean number of injections per month before the pandemic?

2.     The total number of injections in 2021 seems to be higher than that of 2019. The social restrictions and inadequate health care systems may also result in poor glycemic control in DM patients. Is there any data showing the number of newly diagnosed DM or DME during the same period?

3.     The timing of the questionnaire delivered was not mentioned in the article.

4.     The impact of COVID-19 on the visual outcome of DME is missing. 52.2% of the investigated facilities had extended the interval between visits for injections. The effect of changing treatment regimens from prn to TAE or extended prn on the visual outcome of DME is not mentioned in this article.

5.     Line 84-90. The description of how COVID-19 pandemic changes the way of practice in the ophthalmology clinics seems to have little correlation with the number of anti-VEGF treatment and visual outcome of patients with DME.

6.     Authors explained the possible reasons why the COVID-19 pandemic may affect the anti-VEGF treatment in Line 215: The inverse correlation between the increase in the number of COVID-19-positive patients and anti-VEGF treatment may therefore reflect a variety of physical, emotional, and economic anxieties regarding COVID-19. It would be better to show how the anti-VEGF treatment were adjusted for patients who were infected with COVID-19 in Japan. For example, how long were the clinical examinations and treatments delayed in average during the peak of pandemic?

7.     The 6th wave COVID-19 pandemic did not subside at the endpoint of this study. It causes confusion when you tried to explain that the number of monthly anti-VEGF injection returned at the end of this wave of pandemic.

8.     Some parts of english writing are difficult to comprehend. For example: Line 169, Prior to vaccination, the percentages of facilities that responded that the period of time without administering injections was 1 day, 3 days, and 1 week were 4.3%, 13.0%, and 21.7%, respectively…”, and Line 190, as the COVID-19 pandemic has been reported to have reduced the rate of hospital visits greatly [16], possibly because both patients and healthcare workers were concerned about infection.

Reviewer 2 Report

This article explains the impact of COVID-19 pandemic on anti-VEGF therapy for diabetic macular edema in Japan. The number of institutions that was used to study the impact of reduced number of injections seems reasonable, however, it needed to be expanded all clinics around Japan.

Minor comments:

1.      What is the outcome effect on the visual improvement? Any significant decrease in visual acuity was observed in late-stage diabetic patients? Need more explanation in detail.

2.      Since diabetic pateints were more susceptible to die because of COVID-19, did authors consider the number of diabetic patients that died and did not get injection?

3.      Graphs are labeled very small. Please make them larger.
